# Results of a 2021 French National Survey on Management of Patients with Advanced Stage Epithelial Ovarian Cancer

**DOI:** 10.3390/jcm10214829

**Published:** 2021-10-21

**Authors:** Leonor Drouin, Benedetta Guani, Vincent Balaya, Henri Azaïs, Sarah Betrian, Pierre-Adrien Bolze, Yohann Dabi, Yohan Kerbage, Claire Sanson, François Zaccarini, Patrice Mathevet, Fabrice Lécuru, Fréderic Guyon, Cherif Akladios, Sofiane Bendifallah, Elise Deluche

**Affiliations:** 1Department of Gynecology, CHU de Limoges, 87000 Limoges, France; leonor_drouin@hotmail.fr; 2Department of Gynecology and Obstetrics, 1700 Fribourg, Switzerland; benedetta.guani@hotmail.it; 3Department of Gynecology, CHU Vaudois, 1011 Lausanne, Switzerland; vbalaya@hotmail.com (V.B.); Patrice.Mathevet@chuv.ch (P.M.); 4Gynecologic and Breast Oncologic Surgery Department, Georges-Pompidou European Hospital, 75015 Paris, France; henri.azais@aphp.fr; 5Department of Medical Oncology, IUCT Oncopole, 31100 Toulouse, France; betrian.sarah@iuct-oncopole.fr; 6Department of Gynaecologic and Oncologic Surgery and Obstetrics, Centre Hospitalier Universitaire Lyon Sud, Hospices Civils de Lyon, Université Lyon 1, 69100 Villeurbanne, France; pierre-adrien.bolze@chu-lyon.fr; 7Department of Gynecology, Sorbonnes University, Tenon Hospital, AP-HP, 75020 Paris, France; yohann.dabi@gmail.com (Y.D.); sofiane.bendifallah@aphp.fr (S.B.); 8Department of Obstetrics and Gynecology, Lille University Hospital, 59000 Lille, France; yohan.kerbage@chru-lille.fr; 9Surgical Oncology Department, Gustave Roussy Institute, 94805 Villejuif, France; claire.sanson@gustaveroussy.fr (C.S.); francois.zaccarini@gustaveroussy.fr (F.Z.); 10Breast, Gynecology and Reconstructive Surgery Unit, Curie Institute, 75005 Paris, France; fabrice.lecuru@curie.fr; 11Surgical Oncology Department, Bergonié Institute, 33000 Bordeaux, France; f.guyon@bordeaux.unicancer.fr; 12Department of Obstetrics and Gynecology, Strasbourg University Hospital, 67091 Strasbourg, France; cherif.akladios@gmail.com; 13Department of Medical Oncology, CHU de Limoges, 87000 Limoges, France

**Keywords:** ovarian carcinoma, practice management, survey, surgical oncology, medical oncology

## Abstract

Background: The aim of this study was to assess current French practices in the management of patients with advanced epithelial ovarian cancer. Method: a 58-question electronic survey was distributed anonymously to the members of the SFOG (French Society of Gynaecological Oncology), GINECO-ARCAGY (National Investigators Group for Ovarian and Breast Cancer Studies in France) and FRANCOGYN (French research group in oncological and gynaecological surgery). Initial diagnostic workup and staging, pathological data, surgical data, treatments and follow-up strategies were assessed. Results: a total of 107 participants responded to emailed surveys. Most of the respondents were obstetrician-gynaecologists (37.4%), surgical oncologists (34.6%) and medical oncologists (17.8%). According to most (76.8%) participants, less than 50% of patients were eligible for primary debulking surgery (PDS). The LION study criteria were applied in 69.5% of cases during PDS and 39% after chemotherapy. The timing of BRCA testing was very heterogeneous and ranged from 1 to 6 months. The use of bevacizumab as an adjuvant schedule was lower in cases of no residual disease (for 54.5% of respondents) compared to cases of residual disease (for 63.6% of respondents). In cases of BRCA1-2 mutations, olaparib was given by 75.8–84.8% of respondents, whereas niraparib was given in cases of BRCA wild-type diseases. Conclusion: this survey provides an extensive and a unique review of current French practices in the management of patients with advanced epithelial ovarian cancer in 2021.

## 1. Introduction

Ovarian cancer is the eighth most common cancer among women in France. In 2018 there were approximately 5200 new cases and 3500 deaths [1]. The disease occurs mainly in postmenopausal women and is generally diagnosed at an advanced stage (stage IIB-IV) according to the 2014 classification of the International Federation of Gynaecology and Obstetrics (FIGO) [2]. Because of this often late diagnosis, the prognosis remains relatively poor, with 70% relapse within 3 years of the first treatment and 5-year survival of less than 30% of diagnosed patients [3]. Around 90% of these ovarian cancers are epithelial tumours (EOC), in which BRCA mutations appear to be involved in 10–15% of the cases (compared to high-grade cancer) with increased sensitivity to chemotherapy [4].

In recent years, the management of ovarian cancer has become increasingly complex due to surgical and medical advances. The last ESMO (European Society for Medical Oncology)-ESGO (European Society of Gynaecological Oncology) consensus conference on ovarian cancer was held in April 2018 [5] and national guidelines were introduced the following year [6,7]. These recommendations were intended to improve and harmonise the management of patients with EOC and to redefine the role of surgery, chemotherapy and targeted therapy protocols with the use of PARP (poly-ADP ribose polymerase) inhibitors and/or antiangiogenic therapy, follow-up strategies after EOC treatment and management of the recurrent disease. 

In advanced stage EOC, the optimal treatment consists of debulking and chemotherapy that combines platinum and taxanes with or without targeted therapy [5,8]. The objective of the surgery remains macroscopic complete tumour resection, which improves the prognosis of patients in terms of survival [9,10]. The literature reviewed did not reveal any French studies after 2018 that evaluated EOC management practices that followed the latest recommendations. However, prior to 2018, international studies have been conducted addressing specific areas such as surgical techniques [11,12] and general medical practices for advanced EOC [13,14,15,16]. The aim of this study was to assess current French medical and surgical practices in the management of patients with advanced EOC in 2021. 

## 2. Materials and Methods

### 2.1. The Survey

Items of the survey were defined by literature review. The development of the questions was initially requested from practitioners in the SFOG campus (Young of French Society of Gynaecological Oncology) involved in onco-gynaecology. The initial survey questions were formulated in November 2020 after reviewing current practices, recent recommendations and current issues in the field. The initial 108 questions were then modified and revised by different practitioners from the SFOG Campus group and members of the SFOG group. A final national survey of 58 questions was then finalised (Appendix A). 

The survey questions were informed by current evidence relating to different aspects of management of EOC. The questions in the final survey were developed in the French language and included demographic information of participants, pathological/genetic, surgical and medical practices, as well as follow-up strategies in advanced EOC. Questions were mostly formulated with multiple choice answers, but a few items required free text answers. Not all answers were mandatory. Participants had the choice of responding to sections within their field of expertise only. However, the survey had to be fully completed by one party for it to be registered. Therefore, the number of respondents varied among questions.

The final electronic survey assessing the current practice of EOC was emailed to participants during the period from 28 February 2021 to 30 April 2021. The emails were sent through Google forms (https://docs.google.com/forms/u/0/, accessed on 28 February 2021) to all members of the SFOG (French Society of Gynaecological Oncology), SFOG campus (Young of French Society of Gynaecological Oncology), GINECO-ARCAGY (National Investigators Group for Ovarian and Breast Cancer Studies in France) and FRANCOGYN (French research group in oncological and gynaecological surgery). Google forms was used as an online anonymous survey, and no participant-identifiable data were collected. The link to Google Forms was distributed directly by the scientific societies to their members and through the OncoAlert Network. This procedure also guaranteed the anonymity of the participants and precluded us from knowing the exact number of people who were approached. Because responses to the survey were collected anonymously and no personal data were collected, ethical approval was not required.

### 2.2. Statistics 

Data obtained from the survey were analysed by determining the proportions of responses for each question. The captured data from Google sheets were used for descriptive analysis after tabulation. Quantitative data were expressed as means, standard deviations (SD) and extremes. Qualitative data were expressed as a percentage, and the 95% confidence interval (IC) was provided. All statistical analyses were performed with XLStat Biomed software (AddInsoft 2020). 

## 3. Results

### 3.1. Participant Characteristics

A total of 107 participants responded to the emailed survey. Respondents’ characteristics are summarised in Table 1. There were 54 men (50.5%) and 53 women (49.5%), aged between 29 and 64 years. Most of the participants were obstetrician-gynaecologists (37.4%), surgical oncologists (34.6%) and medical oncologists (17.8%). Most of the participants came from university hospitals (40.2%) and comprehensive cancer centres (37.4%). The average length of their professional experience since graduation was 12 years (1 year–37 years). Most of the participants worked in medical facilities, usually involved in the management of EOC. About 44.9% of the participants managed > 50 cancer cases/year and 42.1% treated 20–50 cases/year. About 39.3% had ESGO accreditation. Of the practitioners, 80.3% reported open clinical trials (including early phase trials) in their institutions. More than 10 clinical trials were available for 12.1% of the cases, 5–10 trials for 22.4% of the cases, and 1–5 trials were available for 45.8% of the cases. 

### 3.2. Initial Diagnostic Workup and Staging

Participants were asked about their assessment upon discovery of EOC (Figure 1). A thoraco-abdomino-pelvic CT scan was recommended by a large majority (91.6%) of respondents, followed by pelvic magnetic resonance imaging (MRI) and a positron emission tomography (PET) scan. A CA125 blood test was an essential paraclinical test for 97.2% of responders, whereas an HE4 blood test was less common (26.2%). A third of the participants performed the Risk of Ovarian Malignancy Algorithm (ROMA).

Imaging exams were systematically reviewed by expert radiologists for 78.5% of participants, while preoperative imaging performed outside of the institution was reviewed in only half of the cases. Although 64.5% used Ovarian-Adnexal Reporting and Data System Magnetic Resonance Imaging (O-RADS) in their usual practice, not every centre routinely followed this approach. Of the participants, 103 discussed every patient’s record at tumour board meetings before surgery. The medical specialties included in the tumour board meetings are shown in Figure 2. Finally, patients older than 75 years were systematically referred to a geriatric oncologist in 33.6% of cases and according to their general condition in 65.4% of cases.

Each participant reported the medical specialties participating in the tumour board meetings at their centre. On the x-axis, the percentage of respondents, on the y-axis, the specialties that might be involved

### 3.3. Pathological Data

The majority of patients (82.4%) with rare histological subtypes of ovarian tumours were included in the French national observatory for rare malignant tumours of the ovaries (www.ovaire-rare.org, accessed on 2 October 2021) [17]. The time it took to obtain an oncogenetic consultation varied considerably from one facility to another. Half of the participants provided genetic counselling for their patients within a month. The delays were sometimes longer, ranging from 1 to 3 months for 26.5% of the patients and from 3 to 6 months for 2.9% of patients. The types of BRCA mutations investigated depended on the availability of personal or familial criteria for hereditary cancer. Any delays in obtaining BRCA germline mutation results were respectively reported in Figure 3. A Homologous Recombination Repair Deficiency (HRD) search was performed in 73.5% of cases. The methods used included the Myriad test (87%) and Next Generation Sequencing (NGS) (8.7%) or a combination of both (4.3%).

### 3.4. Surgical Management of EOC

According to the majority of participants (76.8%), less than 50% of patients were eligible for primary debulking surgery (PDS). The decision for PDS was essentially based on the type of surgical procedures required to achieve a complete resection (90.2%). The other criteria included carcinomatosis score (67.1%), global visual impression (26.8%), patient’s age (58.5%) and personal conviction (2.4%) (Figure 4). Assessment of peritoneal carcinomatosis scores was performed by a senior/referred surgeon in 91.5% of cases and one or multiple scores were used: peritoneal carcinomatosis index (PCI) was used in 86.6% of the cases, Fagotti score in 45.1%, Fagotti-modified in 19.5%, and Makar score in 8.5% of the cases. For PDS, exploratory laparoscopy was not performed on the same day of debulking in 75.6% of cases. According to 86.6% of participants, the average interval time between the diagnosis and surgery for patients eligible for PDS was <30 days. 

Cytoreductive surgery was routinely performed by at least two gynaecologic surgeons in 34.1% of cases. Of the 82 surgeons, 79 performed non-gynaecological procedures with help from other surgical specialisations in 60% of cases: 98.7% performed supra-mesocolic surgical procedures, 84.8% performed urological surgical procedures, 51.9% performed diaphragmatic cytoreduction procedures (pleural/cardiophrenic, nodes/supraclavicular, or axillary node resection), and 7.6% performed thoracoscopic procedures.

Concerning lymph nodal investigation 79.3%, 19.5%, and 1.2% of the respondents considered thoraco-abdomino-pelvic CT scan, PET scan, and pelvic MRI, respectively, to be the best imaging modalities for indication of lymph node dissection. The LION (Lymphadenectomy in Ovarian Neoplasms) study criteria were always applied during PDS for 69.5% of respondents. These criteria were applied during interval debulking surgery (IDS) after neoadjuvant chemotherapy (NACT) for only 39% of respondents; 78% of oncological surgeons routinely examined the para-aortic and pelvic nodes area by manual palpation during surgery, while 2.4% reported never doing so (Appendix A).

Hyperthermic intraperitoneal chemotherapy (HIPEC) and pressurised intraperitoneal aerosol chemotherapy (PIPAC) were proposed as general first-line treatment for 41.4% of respondents in their departments (26.8% within clinical trials and 14.6% outside) and, more precisely, during IDS for 37.8% of respondents. Among all the respondents, 70.7% did not perform PIPAC. An ESGO-type standardised operative report was used by only 39% of participants.

### 3.5. Oncology Data

According to 51.5% of responders, the average delay between the first consultation and the first chemotherapy treatment for patients who were not immediately operable was less than 15 days. The neoadjuvant and adjuvant treatment protocols are summarised in Figure 5. For NACT followed by IDS, resecability was assessed after three cycles and accounted for the majority (92.7%) of cases. Paclitaxel (175 mg/m^2^) and Carboplatin AUC5 (every 3 weeks) was the most used combination whether as neoadjuvant (84.8% of respondents), adjuvant after PDS (84.8–78.8% of respondents) or after NACT and IDS (78.8–69.7% of respondents), regardless of tumour and/or germline BRCA1/2 mutations.

The use of bevacizumab (15 mg/kg) (every 3 weeks) as an adjuvant was lower in cases of non-residual disease (CC0) (54.5%) compared with cases of residual disease (CC1) (78.8%). Bevacizumab (15 mg/kg) was not commonly used as neoadjuvant therapy by responders (<10%). Niraparib was used for adjuvant therapy of women without HRD or BRCA mutation (42.2–45.5%). For tumours and/or germline BRCA 1 and 2 mutations, similar results were observed for bevacizumab and the paclitaxel-carboplatin protocol. Olaparib was prescribed as maintenance therapy for patients with partial (84.8% of respondents) or complete response (75.8% of respondents) to first-line chemotherapy in case of the presence of a tumour and/or germline BRCA 1 and 2 mutations. 

### 3.6. Follow-Up Strategies

Clinical monitoring was systematic and essential after the management of ovarian carcinoma. All participants recommended the CA125 blood test as a monitoring tool. Pelvic MRI and PET scans were not reference examinations for surveillance for 96.1% and 81.5% of practitioners, respectively. A thoraco-abdomino-pelvic CT scan was recommended by 69.2% of the participants.

## 4. Discussion

Our survey revealed French practices for the management of advanced EOC in 2021. This study provides the medical community not only with an overview of current practices and the progress made in France but also highlights medical needs. The four major issues highlighted were: the position for primary surgery vs. NACT, the position for lymph node surgery, the choice of therapies (bevacizumab and PARP inhibitors) and oncogenetic delay times.

There were different opinions concerning surgery and neoadjuvant chemotherapy followed by debulking surgery. For many years, upfront PDS has been the standard treatment for EOC. However, randomised controlled trials [18,19,20] and a recent analysis by COCHRANE suggested that there was little or no difference in primary survival outcomes between PDS and NACT [21]. Meta-analysis of two randomised trials (EORTC 55971 and CHORUS) indicated that patients in the NACT group achieved higher CC0 rates and lower perioperative complication rates when compared with patients in the PDS group [22]. However, the CC0 rates in the PDS groups were below 20%, which were lower compared with the rates reported in other studies [23,24]. Moreover, Lyons et al. showed that most patients would benefit from PDS instead of NACT [25]. For practitioners, fewer than 50% of patients had undergone PDS, which was consistent with recent publications [26,27,28,29]. Only one French study indirectly evaluated the rate of PDS and reported a lower rate with 23% of PDS performed [30]. These statements must also take into account the pandemic situation, which may have changed practices at the time the survey was conducted [31]. The French multicentre CURSOC study published in 2021 investigated the distribution of EOC care in France while assessing the proportion of facilities adhering to French quality indicators [32]. Nine health facilities participated in this trial and declared a median number of cytoreduction procedures of 50 per institution per year. This was similar to the data provided by the respondents in our survey. The authors also reported that 530 hospitals in France were accredited for the gynaecological management of ovarian carcinoma, of which 411 treated at least one case surgically in 2018. In addition, in 2018, 3801 patients with EOC underwent debulking surgery in France. Finally, although oncology surgery is not yet a recognised sub-specialty in France, the training of surgeons is nonetheless of crucial importance as the quality of surgery determines the prognosis of patients. A recent French study proposed specific criteria for qualifying facilities as training centres for oncogynaecological surgery as well as for certification of practicing surgeons [33]. The need for training discussed in this article was illustrated by our study in which 60% of respondents required assistance from surgeons of other specialties during cytoreduction surgery.

In the last decade, the prognostic and therapeutic role of lymphadenectomy was widely studied and has often been a source of controversy: some retrospective studies [34] and meta-analyses [35,36] showed improved progression-free survival (PFS) and overall survival (OS) among patients who underwent lymphadenectomy during debulking surgery. However, Pacini’s prospective randomised clinical trial did not show an improvement in OS with lymphadenectomy compared to removal of clinically affected nodes [37]. The last ESMO-ESGO consensus conference [38] on EOC was held in April 2018. The LION study [39], which was not considered in the guidelines, was published in February 2019 [37]. The results of our survey revealed that most participants frequently applied the LION study criteria during PDS. However, a non-negligible number extended the LION criteria to interval surgery, which was not demonstrated in the study because the LION trial did not concern patients after NACT. Similar observations have been reported in other French studies [40].

The 2019 European ESMO [38] and the 2020 NCCN guidelines [8] recommended the use of bevacizumab regardless of BRCA status, in combination with chemotherapy in patients with poor prognosis for a maximum of 15 months, or until disease progression. These recommendations were shared by 78.8% of respondents. A key issue was the use of bevacizumab in complete primary surgery. The questions about its use raised by the ICON7 and GOG218 studies [41,42] were reflected in this survey’s practices. Thus, responders in our study were more likely to prescribe bevacizumab in CC1 (63.6%) than in CC0 (54.5%).

It was interesting to note that practices diligently followed the recommendations concerning PARP inhibitors [38] in cases with BRCA 1 or 2 mutation (germline and/or somatic). Maintenance treatment with olaparib was recommended and niraparib was recommended for non BRCA mutated patients after primary treatment, including surgery and platinum-based first-line therapy [43]. The recent publication of PAOLA-1 has changed practices with the introduction of maintenance treatment with olaparib and bevacizumab [26]. The PAOLA-1 study showed the superiority of the combination of olaparib and bevacizumab over bevacizumab alone in terms of PFS (HR = 0.33; 95% CI (0.25–0.45)) in patients with HRD positive/BRCA mutation tumours [26]. This change in practice was captured by the survey. The management strategy for bevacizumab in combination with niraparib was still under study. The OVARIO study showed very interesting results [44]. The safety of this combination was consistent with the known side effects. Median PFS ab has not been reached. At 6 months, the PFS rate was 89.5% [44].

The inconsistent delays in accessing the prescription for BRCA were considerable. However, the testing was fundamental in patient management [45] and will become increasingly important with the new treatment indications for EOC [26] and also for breast cancer [46], which will contribute to overloading the platforms.

The survey had several limitations. It was constructed in such a way that respondents had the option of answering some sections of the survey, including surgical practices, oncology practices and pathological data, without answering the other ones. Thus, some participants were able to answer questions beyond their field of practice, leading to incomplete data (Appendix A). In addition, our survey was addressed to specialist practitioners who were members of national societies involved in the treatment of gynaecological cancers and in the establishment of national guidelines. Therefore, the practices described in our study did not necessarily reflect those of all practitioners in France. For this reason, our survey probably showed a high rate of HIPEC used as first-line treatment (41.4% of respondents: 26.8% in clinical trials and 14.6% external). Finally, similar to any online survey, this study was subject to selection and reporting bias.

## 5. Conclusions

This survey highlighted contemporary practice and attitudes in the management of patients and provided an overview of the current management of advanced EOC in France in 2021 in relation to international guidelines. To our knowledge, this is the first study to investigate medical practices in oncology, genetics, pathology and surgical practices, and to involve multiple health professionals since the recommendations were introduced in 2018. These practices appear to be in line with current European recommendations. They also provide a basis for further research, including similar surveys extended to include members of international gynaecological cancer societies. In the future, one of the best ways to carry out this type of study would be to create a database for the whole of France, which would be an accurate source for all management strategies.

## Figures and Tables

**Figure 1 jcm-10-04829-f001:**
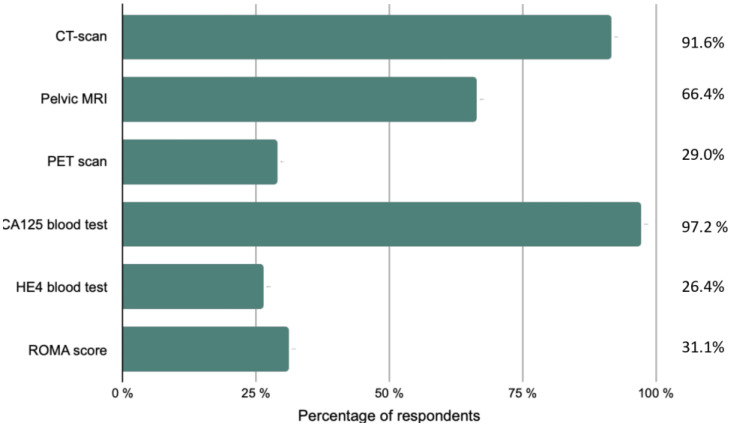
Initial diagnostic workup and staging performed by the respondents.

**Figure 2 jcm-10-04829-f002:**
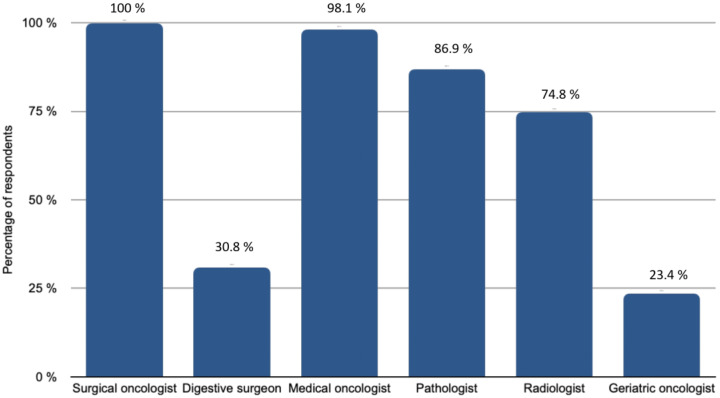
Medical specialities involved in tumour boards.

**Figure 3 jcm-10-04829-f003:**
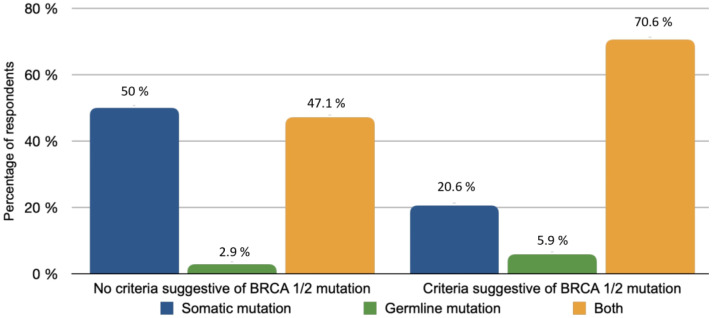
BRCA mutation types based on availability of family and personal criteria.

**Figure 4 jcm-10-04829-f004:**
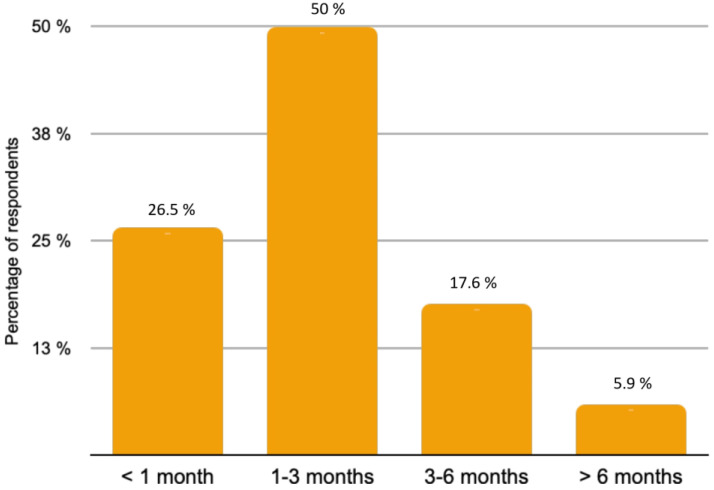
Delay time for obtaining BRCA germline mutation results.

**Figure 5 jcm-10-04829-f005:**
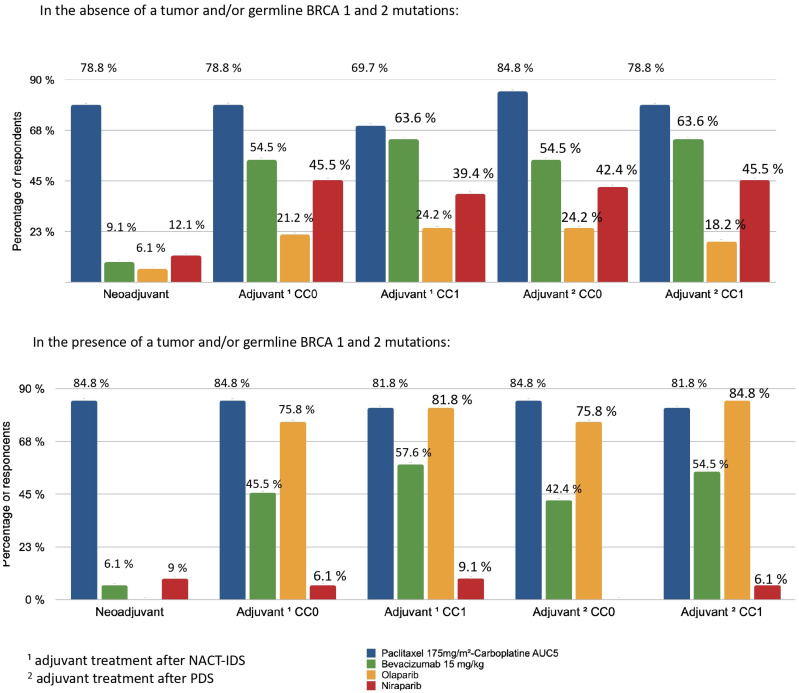
Distribution of treatment indications according to BRCA status and surgery. ^1^ Adjuvant treatment after neoadjuvant chemotherapy (NACT) and interval debulking surgery (IDS); ^2^ adjuvant treatment after primary debulking surgery (PDS).

**Table 1 jcm-10-04829-t001:** Characteristics of survey participants.

Parameters	Participants (*n* = 107)
Sex—no. ^1^ (%)	
Male	54 (50.5)
Female	53 (49.5)
Mean age—years (range)	42.3 (29–64)
Mean professional experience—years (IQR) ^2^	12.1 (5–17)
Medical specialty—no. (%)	
Surgical oncologist	37 (34.6)
Obstetrician-gynecologist	40 (37.4)
Medical oncologist	19 (17.8)
Pathologist	5 (4.6)
Radiation oncologist	1 (0.9)
Geriatric oncologist	1 (0.9)
Radiologist	4 (3.7)
Practice structure—no. (%)	
Private clinic	11 (10.3)
Regional hospital	7 (6.5)
Comprehensive cancer center	40 (37.4)
University hospital	43 (40.2)
Private health establishment of collective interest	6 (5.6)
Ovarian cases managed in the structure—no. (%)	
Less than 20	12 (11.2)
Between 20 and 50	45 (42.1)
More than 50	48 (44.9)
Mean number of surgeons—no. (IQR)	3.6 (2–4)
Clinical trials—no. (%)	
None	11 (10.3)
Between 1 and 5	49 (45.8)
Between 5 and 10	24 (22.4)
More than 10	13 (12.1)
Available resources—no. (%)	
A medical oncology department	101 (94.4)
A surgical oncology department	100 (93.5)
Other surgical specialities	103 (96.3)
A genetic counselor	89 (83.2)
A geriatric oncologist	99 (92.5)
A dietitian or nutritionist	100 (93.5)
A psychologist or psychiatrist	92 (86)
Rehabilitation therapists	86 (80.4)

^1^ Number of respondents; ^2^ Interquartile range.

## Data Availability

Data are available from the corresponding author upon reasonable request.

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
