# Peer review of "Results of a 2021 French National Survey on Management of Patients with Advanced Stage Epithelial Ovarian Cancer"

_jcm, 2021, doi:10.3390/jcm10214829_

Round 1

Reviewer 1 Report

In this manuscript, Drouin et al. explored the clinical management of advanced epithelial ovarian cancer in France through an electronic survey distributed to the members of several national societies. They provided a comprehensive overview of the current practice in France and highlighted potential medical needs.

The manuscript is well written, the approach is methodical and stepwise, the discussion is well presented and balanced, and it provides concise information.

Minor revisions required:

Page 3, lines 112-113  & in other parts of the manuscript: the sentence “Error! Reference source not found” is reported several times throughout the manuscript; to my intending, the figure/table reference should be placed there instead. Please, verify and revise accordingly.

Page 3, line 124, Table 1: there are no data in the table; I can only visualize the total number of participants in the headline, but no data in each specific line in the current form. Please, provide all the missing data.

Page 5, line 144, Figure 2: please, provide a more detailed caption; otherwise, the meaning of the figure could be misleading.

Discussion, page 9, line 295:  the authors assessed that “The OVARIO study showed very interesting results”; can you please discuss further these results?

Supplementary Table S1: the first parameters enlisted are repeated but with different respondents (First line: Answer to surgery questions: 34 … Then: Answer to surgery questions: 83); please, explain this issue or revise the data and the figure accordingly.

Author Response

Reviewer #1

Page 3, lines 112-113  & in other parts of the manuscript: the sentence “Error! Reference source not found” is reported several times throughout the manuscript; to my intending, the figure/table reference should be placed there instead. Please, verify and revise accordingly.

Response : We apologise for the inconvenience and have corrected the problem which must be due to the change of software between the computers.

Page 3, line 124, Table 1: there are no data in the table; I can only visualize the total number of participants in the headline, but no data in each specific line in the current form. Please, provide all the missing data.

Response : We apologise for the inconvenience and have corrected the problem which must be due to the change of software between the computers.

Page 5, line 144, Figure 2: please, provide a more detailed caption; otherwise, the meaning of the figure could be misleading.

Response : To follow reviewer’s comments, we added this sentence,page 5-lin 154: “Each participant reported the medical specialties participating in the tumour board meetings their centre. On the x-axis, the percetnage of respondents, on the y-axis, the specialties that might be involved”

Discussion, page 9, line 295:  the authors assessed that “The OVARIO study showed very interesting results”; can you please discuss further these results?

Response : To follow reviewer’s comments, we added this sentence,page 9 Line 276 : “Safety of this combination was consistent with the known side effects. Median PFS ab has not been reached. At 6 months, the PFS rate was 89.5% [45].” And we added a reference :

Hardesty, M.M.; Krivak, T.; Wright, G.S.; Hamilton, E.; Fleming, E.L.; Gupta, D.; Keeton, E.; Chen, J.; Clements, A.; Gray, H.J.; et al. Phase II OVARIO Study of Niraparib + Bevacizumab Therapy in Advanced Ovarian Cancer Following Front-Line Platinum-Based Chemotherapy with Bevacizumab.; SGO, March 28 2020.

Supplementary Table S1: the first parameters enlisted are repeated but with different respondents (First line: Answer to surgery questions: 34 … Then: Answer to surgery questions: 83); please, explain this issue or revise the data and the figure accordingly.

Response : We have corrected the heading of the first part and added “Answers to pathological questions - no1 (%)”, thanks a lot.

Reviewer 2 Report

In this review paper Drouin et al conducted a survey among 107 onco-gynaecology providers (SFOG) and looked into the management of advanced ovarian carcinoma in France. The electronic survey was sent via emails and utilized the Google forms platform (anonymous participation).

No recent studies pertinent to this project were identified.   Paper: -Error! Reference source no found! in multiple locations. (for Tables and Figures)   Well written study that looks into current practice in France for ovarian carcinomas and can be used in order to improve compliance with the society’s guidelines.    -Would centralization improve compliance and outcomes? (530 hospitals accredited currently..)  - Make cancer society registration compulsory for all cancer centres? Please add comments.   A follow up study for treatment (or center)-specific  patient survival in France would be interesting.

Author Response

Reviewer #2

Paper: -Error! Reference source no found! in multiple locations. (for Tables and Figures)  

Response : We apologise for the inconvenience and have corrected the problem which must be due to the change of software between the computers.

 -Would centralization improve compliance and outcomes? (530 hospitals accredited currently..). Make cancer society registration compulsory for all cancer centres? Please add comments

Response : We appreciate the reviewer’s comments, and have added this sentence in our conclusion page 10, Line 296

“In the future, one of the best ways to carry out this type of study would be to create a database for the whole of France, which would be an accurate source for all management strategies”

Response : We are working within the framework of the SFOG campus to create such a database which will allow in the future to launch large questionnaires on the management of gynaecological cancers. We could even consider qualifying doctors by filling in this database as is the case with EPITHOR.   

A follow up study for treatment (or center)-specific  patient survival in France would be interesting.

We fully agree with this comment and we will see how to implement it in a future project. The creation of a database could also help to answer these questions.
